# Characteristics of Pediatric Primary Healthcare Visits in a University-Based Primary Healthcare Center in Saudi Arabia

**DOI:** 10.3390/children9111743

**Published:** 2022-11-13

**Authors:** Haneen A. Yousef, Moataza M. Abdel Wahab, Shahad Alsheikh, Rizam Alghamdi, Raghad Alghamdi, Najla Alkanaan, Mohammad Al-Qahtani, Waleed H. Albuali, Huda Almakhaita, Mae Aldossari, Abdullah A. Yousef

**Affiliations:** 1College of Medicine, Imam Abdulrahman Bin Faisal University, Dammam 34212, Saudi Arabia; 2Department of Family and Community Medicine, College of Medicine, Imam Abdulrahman Bin Faisal University, Dammam 34212, Saudi Arabia; 3Department of Paediatrics, King Fahd Hospital of the University, Shura Street, Al Aqrabiyah, Al Khobar 34445, Saudi Arabia

**Keywords:** primary healthcare, pediatrics, Saudi Arabia, characteristics, family medicine

## Abstract

This study aimed to identify the characteristics of pediatric primary health care (PHC) visits and evaluate the outcomes of patients presenting with complaints along with their referral and consultation capabilities. This was a retrospective medical record-based study. The study population included any pediatric patient (≤14 years old), including females and males, Saudis, and non-Saudis. Research data were gathered for visits from 2016–2021. Sampling was performed using a stratified random sample based on age groups, followed by simple random sampling with proportional allocation to different age groups. The number of pediatric visits included was 1439 (males, 52.2%). The most common age group was toddlers, and 60% of the total sample was from Saudi Arabia. The most common cause of visits was vaccination (32%), followed by general checkups and/or a well-baby visit (25.4%), and fever (11.2%). Approximately 10% of visits needed referral to other subspecialties. Approximately 50% of visits with complaints concerning ophthalmology, cardiology, and surgical intervention were referred to a specialized department. More awareness needs to be raised about the important role of PHC services in the pediatric age group, as it was capable of handling approximately 90% of their cases.

## 1. Introduction

Family medicine is a specialty that provides comprehensive and coordinated healthcare services to all individuals regardless of age, sex, and impaired organs or body systems. Although Family medicine was recognized in 1969 as the 20th specialty board in the United States, it has only recently been implemented in most Arab countries. However, it remains one of the most widely implemented medical specialties globally [1].

In Saudi Arabia, family medicine, which is practiced at the primary healthcare (PHC) level, is a specialty earned after completing a certificate of training either locally or through international accredited programs. Training includes acquiring competencies in various fields of medicine, including child, adolescent, adult, female, geriatric, and mental health, and providing both acute and chronic care. Moreover, family medicine has contributed to health education and disease prevention [2,3]. In Saudi Arabia, the services provided by family medicine physicians include both children and adults of both sexes. The family medicine scope of services includes both adult and pediatric immunization, short- and long-term management of chronic disease, well-women and baby health, antenatal care and family planning, health promotion, wound care, and urgent care [4].

Some studies conducted in Saudi Arabia revealed that more attention needed to be focused on the various aspects of Family medicine to improve its academic and operational aspects [1]. Furthermore, Saudi Arabia is progressing toward a brighter, more optimistic future, particularly in improving the healthcare system. The National Transformation Program 2020 (NTP 2020), which is part of Saudi Vision 2030, was established to achieve national growth. Healthcare transformation, which is considered one of the themes of NTP 2020, will include PHC as the basis of the health-transformation program and for its critical role in the healthcare system [5,6]. To increase the quality and utilization of healthcare services, Family medicine has been acknowledged as a necessary first-line specialty to refine patients and sort them based on their health needs [7].

In the context of pediatrics, the options for parents seeking medical care for sick children include PHC centers, secondary hospitals, emergency departments, and tertiary hospitals, if needed. Family medicine is not only a bridge to secondary services, but also a family-centered network of services aimed at promoting the healthy development and well-being of patients and their families [8].

Pediatric patients have a different spectrum and frequency of chief complaints [9]. To fully deliver optimized healthcare services to a unique and delicate segment of the population, primary care should be updated regularly to map out and readily tackle different illnesses, identify risk factors, and direct therapy. Very few publications have detailed the frequent presentations of pediatric patients in public hospitals, emergency departments (ED), and PHC visits nationally and internationally. Regarding diverse chief complaints of pediatrics, a recent Saudi retrospective study that analyzed the characteristics of pediatric ED visits showed that the most common complaint was fever. This was followed by respiratory problems, especially in winter, mainly due to viral infections occurring during that period; for example, bronchitis and viral upper respiratory tract infections were superimposed by bacterial infections. In contrast, the summer had the lowest number of visits. Moreover, hematological diseases (e.g., sickle cell disease and other inherited blood disorders) had the highest probabilities of admission both before and after adjustment, followed by endocrine problems, which were statistically significant compared with fever [9]. Since hematological and endocrine diseases require chronic follow-up and assessment, the role of Family medicine is vital for families with children who require continuous care and a variety of other services. To our knowledge, there is a gap in the literature concerning the characteristics of pediatric visits to PHC centers, especially in Saudi Arabia. Thus, we intended to investigate and identify the characteristics of pediatric visits in a university-based PHC center, to describe and assess the most common reasons for the pediatric PHC visits, to evaluate the relation of the season to the pediatric PHC visits, and to evaluate the primary outcomes for the patient’s presenting complaint and the capability (referral, follow-up, discharge) of the PHC centers in dealing with it.

## 2. Materials and Methods

### 2.1. Study Setting and Design

The PHC center of Imam Abdulrahman bin Faisal University (IAU) in the eastern province of Saudi Arabia (SA) conducted this retrospective study based on medical records. The PHC facility at IAU opened its doors in 2016.

University teachers, employees, students, their families, and people in the catchment region were served by this PHC facility, and as per health policy in Saudi Arabia, they received free medical services. Pediatric visits were estimated to account for approximately 10% of the total number of visits to the PHC. The number of staff working in the center was 76, including 29 family medicine physicians. This center featured a laboratory, pharmacy, radiology department, and urgent care services, in addition to family medical clinics. The center is affiliated with the King Fahad University Hospital of the IAU in the Eastern Province of SA, a tertiary hospital consisting of departments of all subspecialties. This is the only referral institution in our PHC. The IAU’s Institutional Review Board committee granted ethical approval (IRB-UGS-2021-01-390). Patient information was kept private and anonymous and was utilized solely for research purposes. Informed consent was not necessary because the data were anonymized.

### 2.2. Study Population

Patient visits were included in the study population. The research data were gathered through visits over the previous 6 years (between 2016 and 2021) and included a pediatric age group ≤14 years old according to our hospital policy (both sexes, Saudis, and non-Saudis).

### 2.3. Inclusion Criteria

-Pediatric age group (≤14 years)-Females and males-Saudis and non-Saudis

### 2.4. Sampling

Initially, we obtained the visit numbers of all visits for 14 years old and below (11,578 visits) and their ages. We then selected a sample of about one-tenth of the visits (1439 visits) by stratified random sampling based on age groups, and then simple random sampling with proportional allocation to the different age groups. The sample size exceeded the minimum required sample size calculated using Epi info ver7.0 to be 1066, assuming a 50% referral of visits, with a margin of error of =3 at a confidence level of 95%.

### 2.5. Data Collection

Data from the sampled medical record numbers (MRN) were obtained by nine data collectors by opening the electronic file of each sampled MRN from university computer devices. The patients’ electronic medical record data were collected using a standardized data collection form. The variables collected were patient date of birth (DOB), sex, nationality, date of the visit (DOV), height, weight, underlying causes of pediatric visits, and chief complaint, as reported by the caregiver and categorized by the involved system, history of the patient, whether consultations or referrals (yes/no) to other departments were needed, and if yes, to which department.

### 2.6. Statistical Analysis

Data were analyzed using IBM SPSS software version 26 and Microsoft Excel. Statistical significance was set at *p* < 0.05. Patient age was calculated as DOV-DOB at the time of visit. Body mass index (BMI) was calculated as weight (kg)/ height (m) squared and then compared to the growth charts of BMI for age [10]. BMI <5th percentile was considered underweight, 5th to <85th percentile as healthy, 85–95th percentile as overweight, and >95th percentile as obese [10]. The time of visits was also categorized into seasons: winter (December, January, February), spring (March, April, May), autumn (September, October, November), and summer (June, July, and August).

The reasons for the children’s visits were based on the International Classification of Diseases (ICD-10) coding system. They were categorized into different categories by consensus among family physicians and pediatric consultants. The categories were determined independently by two family physician consultants with more than 10 years of experience, and then verified by cross-checking each disease condition/diagnosis and its assigned category by two independent pediatric consultants until consensus was obtained on the final categorization presented.

Categories are presented as frequency and percentage. The association between referral and consultation with different potential factors was tested using the chi-square test (Fisher’s exact test was used when the expected frequency was less than 5 in more than 20% of the cells). Moreover, the Z-test of proportion was used to compare categories when needed. Finally, the seasonal variation was tested using the chi-square test to test the difference in percentages among the four seasons.

## 3. Results

### 3.1. Demographic Characteristics

The study sample included 1439 pediatric patient visits concerning 1169 patients. The sociodemographic characteristics of patients are presented in Table 1. The sample consisted of 863 Saudi patients (60%). In addition, 52.2% (n = 751) of the participants were male. Toddlers were the most common age group visiting the family center (28.4%, n= 409), followed by infants (25.4%, n = 366). Neonates were the age group visiting the family center the least (0.8%, n = 11). Normal BMI was observed in 45.8% (n = 460) of our sample, followed by 28.9% (n = 290) for underweight and 1.7% (n = 25) for obesity. The analysis of the data looked at statistical differences in BMI, which was not significant in relation to sex or nationality, with *p* values of 0.681 and 0.631, respectively.

### 3.2. Main Complaints and Involved Systems

The underlying causes of pediatric visits were sorted into categories according to the different systems involved, in addition to those coming for a general checkup and/or well-baby clinic, vaccination, and fever. The most common cause of pediatric visits over was vaccination (32%), which was also the most common cause of the visit for the infant and toddler age groups. The second most common cause of pediatric visits was general checkups and/or well-baby clinics (25.4%), and this was the most common cause of visits for neonates, pre-schoolers, school-aged children, and adolescents. Third, the prevalence of fever was 11.2% (Figure 1).

The highest percentage of neonatal visits (63.6%) was for general checkups and/or well-baby clinics, followed by vaccination (27.3%) and fever (9.1%). Common causes of toddler visits were vaccination (39.6%), general checkup and/or well-baby clinic (31.3%), and fever (9.8%). The pre-school age group had general checkups and/or well-baby clinics as their most common cause of visits (41.5%), followed by vaccination (12.1%) and fever (11.3%). Additionally, the school-age group’s highest causes of visits were for general checkups and/or well-baby clinics (19.1%), followed by fever (17.9%), and gastrointestinal (GI) complaints (9.7%). The common causes for adolescents were general checkups and/or well-baby clinics (22.6%), fever (9.4%), GI complaints (9.4%), endocrine complaints (9.4%), and cardiology complaints (9.4%) (Table 2).

### 3.3. Referral and Causes

Of the total sample, 10% presented with complaints that needed consultations or referrals to other subspecialties. Specifically, 2.8% of the patients only needed a consultation, then they were completely managed and followed up in our PHC center; 1% needed consultation and referral; and the remaining 6.2% were referred and managed in other sub-specialties in the hospital (Figure 2). Of the 95 referred visits, only 7 were referred to the pediatric department, accounting for approximately 0.5% of the total visits. Sixteen visits were referred to the ER, accounting for approximately 1% of the total visits. Approximately 50% of the visits with complaints concerning ophthalmology, cardiology, and surgical intervention were referred to a specialized department (12 referrals of 23 visits, 5 of 10, and 2 of 4, respectively). Only 1.5% of GI complaints and 7.7% of neurological complaints were referred to specialized departments. The percentages for visits to other specialties are shown in detail in Table 3.

The highest referrals were recorded in the adolescent age group (18.9%), followed by the school-age group (16%), and pre-school age group (10.3%; *p* < 0.000). Males recorded a higher number of referrals (12.3%) than females (7.6%; *p* < 0.003). Otherwise, referrals were not associated with any of the studied factors, including nationality (*p* < 0.514), BMI (*p* < 0.351), fever (*p* < 0.362), blood pressure (*p* < 0.457), or seasonal changes (*p* < 0.323).

### 3.4. Seasonal Variation

The underlying causes of pediatric visits, consultation, and referrals were not statistically associated with season (*p* < 0.401, *p* < 0.291, and *p* < 0.863, respectively).

## 4. Discussion

The pediatric population represents a significant segment of the population of Saudi Arabia. In 2020, the General Authority for Statistics in the Kingdom of Saudi Arabia (KSA) estimated the population from age 0 to 14 to be 8.5 million, representing approximately 24.4% of the entire population [11]. Thus, developing and enhancing the care provided to this age group is essential. Though services provided in the pediatric outpatient clinic and ED are essential for pediatric age groups, PHC centers also play a vital role.

Pediatric visits to our PHC center showed that the percentage of visits by males was higher than females. Similarly, two observational cohort studies conducted in 2020 and 2021 in the United States (US) showed that males were more commonly visiting PHC centers [12,13]. According to the General Authority for Statistics in Saudi Arabia, males accounted for 56.8% of the total population, while females accounted for 43.2% in 2021 (mid-year). Specifically, the total number of males in the age group of 0–14 years was higher than that of females in the same age group. The higher percentage of male visits versus female visits could also be due to other reasons that could be explored by qualitative research [11]. Regarding age group, our study revealed that toddlers (1–4 years old) were more frequently observed. However, the two observational cohort studies mentioned earlier showed that those aged 12–17 years, which is equivalent to school-age and adolescent age according to the classification used in our study, and the age group from 5–12, respectively, were the most common patients seen in PHC centers [12,13]. The Saudi PHC system works for all pediatric age groups, including neonates, infants, toddlers, pre-school, school age children, and adolescents. The differences in the frequency of presentations of different age groups might be explained by the mandatory vaccination program in Saudi Arabia, where most of the vaccines are scheduled for this specific age group (1–4 years old). According to the Saudi national immunization schedule, nine mandatory vaccines are already required by the age of 4 years, contributing to a higher frequency of visits in this age group.

An unexplained variation was noted between different studies regarding demographic factors, and as stated earlier, the tremendous gap in the literature has contributed to this issue; thus, more studies are required to develop a stronger insight. The seasonal impact on visits was insignificant in this study. However, compared with the ER Saudi retrospective study, winter had the highest number of presentations, and this was commonly due to respiratory infections [9]. The insignificance of seasonal variation that was found in our study could be attributed to the fact that most of the visits were for vaccination followed by general check-ups and/or well-baby clinics, which are not affected by weather changes.

Al-Agha et al. (2020) revealed that underweight percentage was less than that reported in Jeddah 2020 [14]. The percentage in males was less than that reported in Qatar, comparable to Palestine, Syria, and UAE, and more than that previously reported in KSA, Morocco, Oman, Sudan, Tunisia, Turkey, and Yemen. The percentage of females with obesity was comparable to that reported previously in KSA, Oman, Palestine, Qatar, Syria, UAE, and higher than that in Sudan, Tunisia, Turkey, and Yemen [15].

The most common causes of pediatric visits to our PHC center were vaccination (32%), general checkup and/or well-baby clinic (25.4%), fever (11.2%), dermatology (4.7%), and ENT (4.7%). However, British research on common children’s illnesses and the frequency of visits to general practitioners was published in January 1998. The parents of 1805 children kept a health journal for three weeks during the research period, noting their children’s symptoms and doctor visits. The study reported ear symptoms as the most common complaint under ENT (36 %), followed by dermatology (28 %) [16]. The high percentage of vaccinations, general check-ups, and well-baby clinic visits in our center can be attributed to the comprehensive role of family medicine in Saudi Arabia in preventive and curative services, as stated earlier. In addition to its role in health promotion, it provides vaccinations and serves as an entry point for managing diseases and symptoms [1,4]. Besides vaccination and well-baby clinic visits, ENT and dermatology complaints were also frequent in the previous study. Hence, effort is required to enhance the knowledge and skills of general practitioners and family medicine physicians in dealing with such cases.

Fever is considered a frequent presenting complaint and one of the most common reasons parents seek medical advice [17]. Although fever can be associated with different types of infection, most cases are considered non-urgent, self-limiting, and do not require treatment. Fever is a complaint that causes a considerable workload in primary care centers [18]. As mentioned previously, it was the third most common cause of pediatric visits to our center. We compared this finding to that of a recent Saudi retrospective publication that studied the common pediatric presentations in the ED of our receiving tertiary hospital and found that fever cases were the most common cause of presentations. The results also showed that many emergency cases were not urgent and could have been treated by primary care physicians [9]. The lack of urgency in the ED cases reflects the underutilization of primary care services by the Saudi population. Therefore, more efforts are required to educate the public on the services provided by PHC centers to increase their utilization, especially for the pediatric age group.

The aim of PHC is to serve as an entry point for healthcare systems [19]. As a result, referrals to secondary care are expected in a few cases. A retrospective Saudi study analyzed the pattern of referral from PHC to secondary care. The study found that the total number of referrals from PHC was 3.6%, with patients mostly referred to general surgery, ophthalmology, and obstetrics and gynecology [20]. Another study conducted in the US examined the referral patterns of pediatric patients from PHC to other specialists. The result revealed that total referrals were estimated to be 2.29%. Ear, nose, and throat (ENT), orthopedic surgery, and ophthalmology constitute the most common surgical departments regarding the number of referred cases, and dermatology, allergy, and radiology constitute the most common medical departments [21]. However, our study estimated a total referral rate of 7.2%. The higher referral rate in our center could be due to the close connection between our center and the receiving tertiary hospitals. The combined system of our PHC and the receiving tertiary hospital helped ease referrals without the need for medical reports. In addition, most of these referrals were surgery-related, and our center did not provide some of the required procedures.

Approximately 50% of visits with complaints concerning ophthalmology, cardiology, and surgical intervention were referred to the specialized department (12 referrals of 23 visits, 5 of 10, and 2 of 4, respectively). Regarding all cardiology referrals, the documented reason for referrals was systolic murmurs. Heart murmurs are frequently encountered in the PHC setting [22]. In a study that evaluated the factors prompting referral of children with heart murmurs to pediatric cardiology, 61% of the murmurs were found to be functional or innocent. The investigators hypothesized that to reduce unnecessary anxiety and referrals, increased education of healthcare providers and parents is required [23]. An approach that helps busy clinicians identify murmurs that need referral was addressed in a review article to help family physicians rule out pathological murmurs by performing a complete, organized cardiac examination [24,25].

As for the ophthalmology referrals, all cases were due to esotropia. The opposite was observed in Qassim Province, Saudi Arabia, where children under the age of 18 make up 50% of the overall population in the studied region. Primary school children had a strabismus prevalence of 5.8% and an amblyopia prevalence of 3.9%. Additionally, fewer referrals than anticipated have been made for the treatment of refractive error, particularly in children. Uncorrected refractive error, particularly in school-aged children, is one of the main causes of visual impairment worldwide. The World Health Organization recommends screening for eyesight and refractive error in 7th and 10th grade pupils as a result. Children in Riyadh’s secondary and preparatory schools had rates of up to 44.5% and 43.2%, respectively [26].

The evident underutilization of PHC by pediatrics raises the question of the factors influencing parents’ health-seeking behavior. A recent systemic review addressed the factors influencing parents’ decisions on unscheduled pediatric visits to the ED and out-of-hours PHC services in many countries. The study identified the following factors: need for reassurance; perceived severity and urgency of the condition; higher accessibility and low waiting time in the ED; and public perception of a better quality of health services in the ED. A systemic review also recognized a few specific parental factors; for instance, depressed mothers are more likely to take their children to the ED than PHC [27]. Younger parents tend to seek ED health services more often [27,28]. A Riyadh-based study investigated these influencing factors. Upon asking parents, 76% stated that if they knew that PHC could deal with their case, they would not seek the ED. Moreover, 80% answered with “yes” when asked if the ED was not their only available option; the same group also showed that 40% of them would prefer phone counseling, 20% would prefer PHC, 17% would prefer outpatient clinic, and 69% would prefer a pharmacy if the ED were not available. Although 90% of the study population knew about PHC, with 76% having an available PHC in their neighborhood, only 24% knew that PHC had an ED [29]. Therefore, knowing the factors influencing parents’ decisions highlights the need for public educational programs focusing on the role of family physicians and PHC.

Despite significant gains in Saudi Arabia’s healthcare industry over the last few decades, the country’s PHC system has faced several issues. The challenges include increased demand due to rapid population growth, high healthcare costs, inequitable access, concerns about care quality and safety, a growing burden of chronic diseases, a less-than-effective electronic health system, suboptimal cooperation, and coordination between other sectors of care, and a highly centralized structure. A Saudi-based study conducted in 2021 examined the capacity of 2319 PHC centers in Saudi Arabia and showed that 24/7 services are infrequent, paper-based documentation is still standard, and emergency services are less likely to be served [5]. These challenges have accompanied the recent introduction of family medicine in Saudi Arabia. Thus, further development must be implemented to overcome these challenges and provide better care for the pediatric age group and their families.

One of the limitations of this study was the sample size, which was affected by technical issues. It could have been better to take the whole sample, but we had to take a smaller sample of visits and adjust for it statistically.

## 5. Conclusions

The findings of the study concluded that PHC can manage approximately 90% of pediatric visits, and the most common underlying causes were vaccination, general checkup and/or well-baby clinic, and fever. The study findings highlight the need to increase awareness of the vital role of PHC services in the pediatric age group, aid in the development of primary care guidelines, resource allocation, and the design of training programs and curricula that are based on Saudi pediatric population characteristics. In addition, it emphasizes the necessity for larger-scale primary care studies and serves as a call for primary care researchers globally to investigate the prevalent pediatric conditions in their respective regions.

## Figures and Tables

**Figure 1 children-09-01743-f001:**
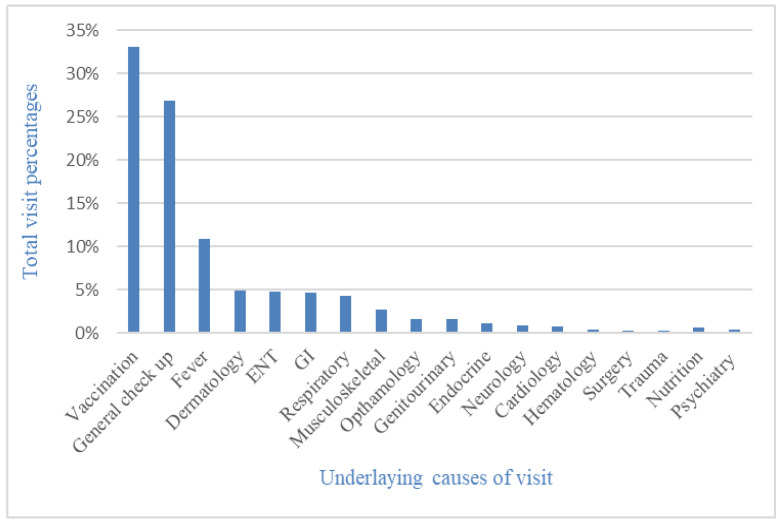
Percentages of systems involved in main complaints of pediatric visits. ENT, ear nose and throat; GI, gastrointestinal.

**Figure 2 children-09-01743-f002:**
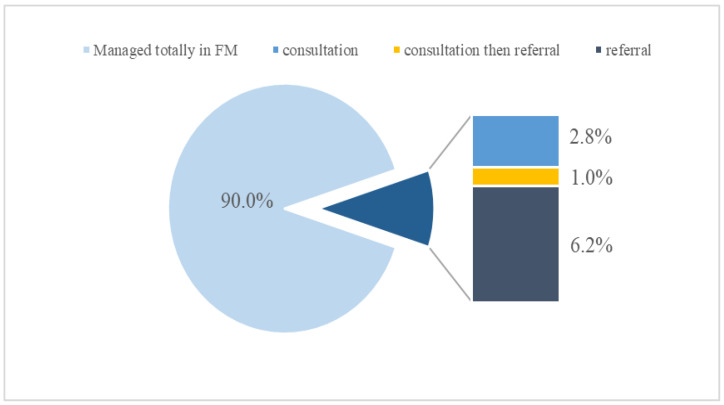
Distribution of visits, consultations, and referrals. FM, family medicine.

**Table 1 children-09-01743-t001:** Sociodemographic characteristics of pediatric visits to PHC center during 2016–2022.

Sociodemographic Characteristics		n	%
Age	Neonate (<29 days)	11	0.8%
	Infants (29 d to <1 year)	366	25.4%
	Toddler (1 to <4 years)	409	28.4%
	Pre-school (4 to <7 years)	282	19.6%
	School (7 to <13 years)	318	22.1%
	Adolescent (13–14 years)	53	3.7%
Sex	Male	751	52.2%
	Female	688	47.8%
Nationality	Saudi	863	60.0%
	Non-Saudi	576	40.0%
	Underweight	290	28.9%
BMI	Normal	460	45.8%
	Overweight	115	11.4%
	Obese	140	13.9%

n, number; BMI, body mass index.

**Table 2 children-09-01743-t002:** Systems involved in main complaints of pediatric visits to PHC according to age.

Systems Involved in Pediatrics Visits	Neonates (<29 Days) n = 11		Infants (29 d to <1 Year) n = 366		Toddler (1 to <4 Years) n = 411		Pre-School (4 to <7 Years) n = 278		School Age (7 to <13 Years) n = 275		Adolescent (13–14 Years) n = 53		Total n = 1395	
	n	%	n	%	n	%	n	%	n	%	n	%	n	%
Vaccination	3	27.3%	250	68.3%	162	39.6%	34	12.1%	8	2.5%	4	7.5%	461	32.0%
General checkup/ Well-baby check up	7	63.6%	49	13.4%	127	31.3%	117	41.5%	62	19.1%	12	22.6%	375	25.4%
Fever	1	9.1%	17	4.6%	40	9.8%	32	11.3%	57	17.9%	4	7.5%	151	11.2%
Dermatology	-	-	16	4.4%	9	2.2%	12	4.3%	26	8.2%	5	9.4%	68	4.7%
ENT	-	-	8	2.2%	29	7.1%	14	5.0%	15	4.7%	1	1.9%	67	4.7%
GI	-	-	6	1.6%	11	2.7%	12	4.3%	31	9.7%	5	9.4%	65	4.5%
Respiratory	-	-	9	2.5%	12	2.9%	22	7.8%	14	4.4%	3	5.7%	60	4.2%
Musculoskeletal	-	-	3	0.8%	9	2.2%	7	2.5%	16	5.0%	2	3.8%	37	2.6%
Opthalmology	-	-	3	0.8%	5	1.2%	4	1.4%	11	3.5%	-	-	23	1.6%
Genitourinary	-	-	2	0.5%	4	1.0%	7	2.5%	8	2.50%	2	3.8%	23	1.7%
Endocrine	-	-	-	-	1	0.20%	4	1.4%	7	2.2%	5	9.4%	15	1.0%
Neurology	-	-	1	0.3%	-	-	1	0.4%	7	2.2%	2	3.8%	13	1.0%
Cardiology	-	-	-	-	-	-	2	0.7%	3	0.9%	5	9.4%	10	0.7%
Hematology	-	-	-	-	1	0.2%	5	1.8%	-	-	-	-	6	0.4%
Surgery	-	-	2	0.5%	-	-	1	0.4%	-	-	1	1.9%	4	0.3%
Trauma	-	-	-	-	-	-	2	0.7%	2	0.6%	-	-	4	0.3%
Nutrition	-	-	-	-	-	-	2	0.8%	5	1.60%	1	1.90%	8	0.6%
Psychiatry	-	-	-	-	1	0.2%	-	-	3	0.90%	1	1.9%	5	0.4%

n, Number; PHC, primary health care; ENT, ear nose and throat; GI, gastrointestinal.

**Table 3 children-09-01743-t003:** Consultation or referral from the PHC center to other specialty departments in the hospital.

Department	Visits (n)	Referrals	Percentage
**Pediatric:** gastroenteritis, milk allergy, dietary modifications, congenital forehead mass, congenital sternal anomaly.	From total visits (n = 1439)	7	0.48%
**ED:** dehydration, fever, head trauma, foot trauma, unstable vital signs,		16	1.11%
other specialized departments			
**ENT:** cochlear implant clinic, laryngomalacia, nasal septal deviation, tonsillectomy, hearing impairment, speech and audiology assessment, stuttering, snoring	67	21	31.34%
**Ophthalmology:** esotropia	23	12	52.17%
**Orthopedic:** flat foot, fracture, left radial osteochondroma, right wrist trauma, right wrist fracture, scoliosis	41	11	26.83%
**Dermatology:** atopic dermatitis, cryotherapy, skin hyperpigmentation	68	9	13.24%
**Cardiology:** systolic murmur	10	5	50.00%
**Endocrine**	15	4	26.67%
**Genitourinary:** empty scrotal sac, undescended testis	23	3	13.04%
**Surgery:** right inguinal area swelling, suture removal	4	2	50.00%
**Hematology:** elevated PTT	6	2	33.33%
**Dental**	1	1	100.00%
**GI**	65	1	1.54%
**Neurology**	13	1	7.69%

ER, emergency department; ENT, ear, nose and throat; GI, gastrointestinal.

## Data Availability

The data that support the findings of this study are available upon request from the corresponding author.

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
