# Peer review of "Characteristics of Pediatric Primary Healthcare Visits in a University-Based Primary Healthcare Center in Saudi Arabia"

_children, 2022, doi:10.3390/children9111743_

Round 1
Reviewer 1 Report
The paper by Yousef et al. examines the pediatric population visiting primary health care facilities in Saudi Arabia.
Some addition information would be appreciated to better understanding the data.
1) What percentage of all patients visiting these facilities are children?
2) What are the other options for parents seeking medical care for sick children?
3) Are these visits to PHC clinics commonly covered by health insurance?
4) Could you briefly comment on the role of primary care pediatricians in Saudi Arabia?
5) In table 1, 28.9% of children were underweight (<fifth percentile), and 13.9% were obese (>95thpercentile). These numbers are alarming! Do you have data for your region to compare this with? Did you analyze the statistical association of BMI-percentile and gender or nationality?
Minor Points:
Line 87-88: What is the difference between a family medicine consultant and a family medicine specialist?
3.1., Demographic characteristics, do you have data on the nationality of non-Saudi children?
3.1., is the difference in gender (males/females) statistically significant?
3.3 Referral and causes: The low percentage of children referred to a pediatric hospital is surprising (only 6.32% of those who were referred). Could you comment on why this rate is so low?
Please check the use of capital letters and brackets.
Thank you for letting me review your manuscript.
Reviewer 2 Report
The manuscript describes a medical record review of reasons for pediatric patient visits to a PHC entity in Saudia Arabia. The main drawback of the article is that it is unclear where the PHC entity sits in the continuum of care resources. In addition, while the authors describe this is Family Medicine, the care described seems to be more akin to primary care pediatric care in many parts of the world, since Family Medicine is primary care that serves both children and adults. Framing the health care system more clearly would be helpful in understanding the usefulness of these data. Are PHC organizations designed to provide primary health care, while emergency care and specialty care are provided through other entities? Without understanding the overall range of health resources in the location where the study was conducted, it is hard to interpret the meaning of the data. In addition, comparisons to other health care systems are not relevant. For example, one comparison to the US indicates high child ER use, but in the US this is dictated by lack of insurance, language barriers etc. Thus the discussion is confusing (lines 207-2018) and references to lack of training, lack of resources, and need for circumcision (line 247) need better explanations. A few other small points: The description refers to FM consultant and FM specialists-what is the difference? Was only 1 visit per individual patient included in the sample? Clarify whether 14 year olds were included or excluded in the study. The text says <14 years but the demographics provide data for 13-14 year olds. This age range is also different from traditional pediatric practices in other parts of the world which do extend through adolescence, although sometimes there is a specialty in adolescent medicine.
Round 2
Reviewer 2 Report
The authors have made some improvements in this manuscript. However, it still would benefit by better use of citations and a stronger discussion relevant to development of the Saudi health care system for primary pediatrics. First, while it appears that the focus is on primary care visits for children 14 or under seen by family medicine physicians, and a broad definition of family medicine is provided, the manuscript seems to leave adult care from the list (lines 41-43). This is important perhaps in the ways the population might be utilizing FM for pediatric care. Here in the US a sizeable portion of the pediatric population uses pediatric primary care while FM tends to be more adult care, especially in urban areas. Second, the authors attempt to compare the system with other countries, but several of the cited articles are designed for different purposes than describing pediatric primary care, so comparisons remain confusing. For example, citation #7 was written to describe robust services for children with special needs, not the general pediatric population. It should also be noted that a long sentence/phrase is directly copied from this article in lines 69-72 which needs to be put in quotation marks and further cited or else eliminated as not relevant. Third, it is confusing to compare the manuscript's data to other studies since the labels for presenting problem or complaint in Figure 1 and Table 2 are a list of medical specialties, not complaints. Table 3 should also provide an n for the number of visits--it this the 10% of the 1439 total visits sampled that received referrals, or about 140 cases? Further in comparing the results to other studies in the discussion, the authors remark about the higher percentage of male visits vs. female- although the cited study shows a much closer proportion that could be explained by the natural occurring phenomenon that slightly more males are born than females. However, there could also be cultural reasons why families seek medical care more frequently for males (e.g., male babies can be encouraged to be more active than female babies and therefore exposed to more illness or accidents). In terms of the distribution of ages seen in pediatric primary care (citation 11), what is more interesting is that more than half of the visits in this paper were for children age 3 or under, rather than that in the US a higher proportion of children 6-17 are seen in care. That would seem to be an issue of lack of access to care or views about the lack of need for care for children after toddlerhood that should be discussed. Perhaps the Saudi system works well for young children but not for school age children? In addition, the comparison to Cambodia is also inaccurate (citation 16) since that manuscript specifically talks about visits that are characterized as emergency type, unscheduled hospital visits and in the context of developing an emergency medicine specialty in Cambodia. It is then confusing then the authors indicate in line 350 that PHC that is the focus of this paper has an emergency department or emergency services as part of the care. Lines 350-352 in the discussion thus need clarification. So overall, the authors could do a much better job explaining their own data and describing what that might mean for their own health care system. Finally, there are a number of sentences that are incomplete, lack subject verb agreement, or have capitalization inserted where it does not belong. A thorough English language edit needs to be conducted.
Author Response
|
|
Comments |
Response |
|
1 |
However, it still would benefit by better use of citations and a stronger discussion relevant to development of the Saudi health care system for primary pediatrics. First, while it appears that the focus is on primary care visits for children 14 or under seen by family medicine physicians, and a broad definition of family medicine is provided, the manuscript seems to leave adult care from the list (lines 41-43). This is important perhaps in the ways the population might be utilizing FM for pediatric care. Here in the US a sizeable portion of the pediatric population uses pediatric primary care while FM tends to be more adult care, especially in urban areas. |
Paragraph rewritten
|
|
2 |
The authors attempt to compare the system with other countries, but several of the cited articles are designed for different purposes than describing pediatric primary care, so comparisons remain confusing. For example, citation #7 was written to describe robust services for children with special needs, not the general pediatric population. It should also be noted that a long sentence/phrase is directly copied from this article in lines 69-72 which needs to be put in quotation marks and further cited or else eliminated as not relevant. |
Paragraph edited
|
|
3 |
it is confusing to compare the manuscript's data to other studies since the labels for presenting problem or complaint in Figure 1 and Table 2 are a list of medical specialties, not complaints. |
|
|
4 |
Table 3 should also provide an n for the number of visits--it this the 10% of the 1439 total visits sampled that received referrals, or about 140 cases?
|
Table Edited and new information is added to abstract, results, and discussion.
|
|
5 |
Further in comparing the results to other studies in the discussion, the authors remark about the higher percentage of male visits vs. female- although the cited study shows a much closer proportion that could be explained by the natural occurring phenomenon that slightly more males are born than females. However, there could also be cultural reasons why families seek medical care more frequently for males (e.g., male babies can be encouraged to be more active than female babies and therefore exposed to more illness or accidents). |
Added to the discussion:
|
|
6 |
In terms of the distribution of ages seen in pediatric primary care (citation 11), what is more interesting is that more than half of the visits in this paper were for children age 3 or under, rather than that in the US a higher proportion of children 6-17 are seen in care. That would seem to be an issue of lack of access to care or views about the lack of need for care for children after toddlerhood that should be discussed. Perhaps the Saudi system works well for young children but not for school age children?
|
Paragraph added to discussion
|
|
7 |
In addition, the comparison to Cambodia is also inaccurate (citation 16) since that manuscript specifically talks about visits that are characterized as emergency type, unscheduled hospital visits and in the context of developing an emergency medicine specialty in Cambodia. It is then confusing then the authors indicate in line 350 that PHC that is the focus of this paper has an emergency department or emergency services as part of the care. Lines 350-352 in the discussion thus need clarification. So overall, the authors could do a much better job explaining their own data and describing what that might mean for their own health care system. |
Paragraph rewritten
|
|
8 |
Finally, there are a number of sentences that are incomplete, lack subject verb agreement, or have capitalization inserted where it does not belong. A thorough English language edit needs to be conducted. |
|

Round 3
Reviewer 2 Report
The authors continue to make improvements, but some of the additions made raise further questions. First, on p. 1-2 the description of family medicine's scope and comparison between SA and the US is somewhat imprecise. US Family Medicine practice by definition encompasses most of that listed as provided in SA PHC. However, there are some confusing topics such as listing dental care and occupational care. The reference citations seem to indicate multidisciplinary primary care clinics that may include such services as dental care, but it's not clear that FM clinicians provide dental care. Also later in the manuscript it is indicated that parents don't know that PHC provides ER care. This again is confusing; do the authors mean urgent care? It is common in the US that primary care does deliver urgent care sessions but not emergency room care. The imprecision in describing the health care entity that is the focus of the manuscript is a problem. Further, it is noticed that while the authors indicate the data are visit based and not patient based in their response to reviewers, this was not added to the manuscript. This is a problem since they make much of the fact that the distribution of visits is skewed toward children under 4 because of well-baby and vaccination visits. This would be true for any pediatric population but with the small sample in this study (1459 visits vs cited studies of millions of child visits in primary care), multiple visits over the 4-year data collection period by neonates growing into toddlerhood could substantially skew the results. This also then leads to the discussion lines 348-353 where much is made of the frequency of younger child visits which indeed are likely due to the common practice for all primary child health care to have more frequent infant and toddler visits. It might help gain perspective on how many cases are reflected in the visit data presented to get a better understanding of the population distribution. At the very least this needs to be mentioned as a major limitation when the sample size is so small. Also there is a citation to distribution of child visits to illness care (#17) that reflects parent views, not coded medical records, and this citation on purpose samples only for sick visits, so comparing it to all visits in the manuscript doesn't make sense.
The authors' attempt to clarify Table 3 leaves it in an edited form where lines are crossed out and others added which makes it hard to follow. Also, it seems some of the percent referral numbers have the decimal point in the wrong place, inflating the number.
Another problematic area in the revision is the discussion of emergency room use. The references are not particularly relevant and understanding this part if the manuscript is unclear as noted earlier, since the authors refer to PHC having an ER/ED. Do they really mean this, or are they talking about urgent care--which is common in most primary health care around the world--and for children, fever would indeed be the most common reason for parents seeking care, although the fever could be due to any number of types of infections. In this section citation #30 is a study of adults not children; citation #31 is a protocol only article without data; and citations 32 and 33 were not able to be accessed. I think the authors can make their point without misrepresenting some of the citations.
Finally, it is fine to emphasize which specialties required more referrals, but when the numbers are so small, making much out of 50% being for cardiology (out of 10 visits) or ophthalmology (23 visits) seems to reinforce that PHC can provide most of children's care. Also a note on Table 3-the percent referral for ENT (21 out of 67 cases) seems to be wrong.
There also remain some editing issues.
